# Identification of the Risk Factors Associated with Low Bone Density in Peri- and Early Postmenopausal Women

**Dave B. Patel, Briana M. Nosal, Manije Darooghegi Mofrad and Ock K. Chun \*** 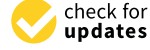

Department of Nutritional Sciences, University of Connecticut, Storrs, CT 06269, USA;
briana.nosal@uconn.edu (B.M.N.); manije.darooghegi_mofrad@uconn.edu (M.D.M.)
\* Correspondence: ock.chun@uconn.edu

**Abstract:** Evidence has shown that one of the most prevalent chronic conditions in postmenopausal women is osteoporosis. Despite the development of some medications, there are still safety and adherence concerns, and, thus, attention has been placed on understanding modifiable risk factors for bone loss. This study aimed to examine the differences in various sociodemographic and body composition factors, physical activity components, and nutrient and food group intake levels among peri- and early postmenopausal women with whole-body bone mineral density (BMD) Z-scores greater than and less than zero. This cross-sectional study utilized baseline data obtained from 45 peri- and early postmenopausal women aged 45–60 years old who participated in a 6-month three-arm, randomized, double-blind, placebo-controlled clinical trial that evaluated the effects of blackcurrant supplementation on bone metabolism. Anthropometric data, BMD values obtained via DXA scans, and self-reported demographic, health, dietary, and physical activity data were collected at baseline. Overall, participants with whole-body BMD Z-scores greater than zero had higher body mass indexes (BMIs), lean mass, fat mass, android fat percentages, ratios of trunk fat mass to limb fat mass, resting metabolic rates, relative skeletal muscle indexes, total and occupational physical activity, alcohol intakes, trans fatty acid intakes, and adequacy of potassium intake, but lower adequacy of vitamin E ($p < 0.05$). In addition, total calorie intake was positively correlated with added sugar, vitamin E, potassium, alcohol, trans fatty acids, calcium, and vitamin D intakes ($p < 0.05$); whole-body BMD was positively correlated with added sugars ($p < 0.05$); vitamin D intake was positively correlated with vitamin E, potassium, trans fatty acid, and calcium intakes ($p < 0.05$); and calcium intake was positively correlated with added sugar, vitamin E, potassium, and trans fatty acid intakes ($p < 0.05$). These findings suggest that numerous body composition factors, components of physical activity, and dietary factors are related to bone health in adult women in the menopause transition.

**Keywords:** diet; body composition; physical activity; bone mineral density; women; menopause



## 1. Introduction

Osteoporosis is one of the most prevalent chronic conditions that postmenopausal women face, with an estimated prevalence of 30% in postmenopausal women [1]. It is estimated that there are 9.1 million women and 2.8 million men with osteoporosis. Women have a higher risk of developing osteoporosis due to changes in hormonal balance and increased levels of bone resorption with postmenopausal status. Women also face a higher risk of bone fractures due to osteoporosis. As many women live for over 20 years after menopause diagnosis, these bone injuries can significantly affect an individual's quality of life [2].

Osteoporosis can be diagnosed by determining the bone mineral density (BMD) of different regions throughout the body, such as the hip and lumbar spine due to high risks for fracture, using a dual X-ray absorptiometry (DXA) scan [3]. The World Health Organization (WHO) defines osteoporosis in postmenopausal women as a BMD T-score in the spine, hip, or forearm of −2.5 or less [3]. The definition of osteoporosis in premenopausal women, in

the absence of fragility fracture(s), is a BMD Z-score of $\leq -2.0$ [4]. Despite the development of some medications, there are still safety and adherence concerns [5,6]. Thus, identifying modifiable risk factors for the prevention of decreased BMD may aid in determining effective strategies to prevent osteoporosis in adult women in the menopause transition.

Previous reports indicate that bone health has been found to be associated with multiple body composition factors in postmenopausal women, including higher BMI [7,8], resting metabolic rate (RMR) [9], android fat [10], and higher trunk fat mass versus limb fat mass [11], associated with higher BMD in postmenopausal women. Additionally, some studies found an association between decreased lean and fat mass and skeletal muscle mass with lower BMD and osteoporosis in postmenopausal women [9,12]. The previous literature has supported that physical activity can prevent loss of bone density and increase BMD in postmenopausal women [13–16]. However, research is still limited on specific physical activity regimens for postmenopausal osteoporosis prevention, and the findings of many studies vary [13–15].

Some focus has been placed on assessing the effectiveness of nutrients and dietary patterns on postmenopausal osteoporosis risk. Studies have found some effectiveness of supplementation with calcium and vitamin D in reducing bone loss in postmenopausal women [17]. Several studies have assessed dietary patterns and specific food group consumption for supporting bone health in postmenopausal women, including dairy intake [18–20], oily fish [20], various macro- and micronutrients (e.g., fats, protein, vitamin C, iron, potassium, and zinc) [21], tea consumption [22], and fruits and vegetables [23]. Research has suggested that some dietary components are associated with increased BMD loss, such as coffee [22], polyunsaturated and monounsaturated fatty acids, retinol, and vitamin E [23]. Associations have also been studied regarding socioeconomic status and BMD in postmenopausal women and significantly higher socioeconomic levels have been found among people with greater BMD [24].

Due to the prevalence of osteoporosis among postmenopausal women and limited treatment options, it is essential to understand factors that may prevent the risk of osteoporosis and bone loss. However, many studies have reported conflicting results, and there is limited research identifying how physical activity is associated with bone loss in this population. In addition, much of the previous research findings focus on factors associated with osteoporosis in postmenopausal women. However, there is a lack of research focusing specifically on women in the menopausal transition and how this may differ from findings in general postmenopausal women. Thus, the aim of this study was to investigate how various sociodemographic, body composition, physical activity, and dietary factors are associated with low BMD in peri- and early postmenopausal women.

## 2. Materials and Methods

### 2.1. Study Design

This cross-sectional study was conducted on 45 peri- and early postmenopausal women aged 45–60. Participant data in the present study were collected during the baseline visits of a clinical trial investigating the effect of blackcurrant supplementation on mitigating bone loss in peri- and early postmenopausal women [25]. Women were recruited from Northeastern Connecticut, and the inclusion criteria for the trial included women (1) aged 45–60 years old, (2) maintaining a normal exercise level of less than 7 h per week, (3) willing to consume a dietary blackcurrant supplement or placebo, (4) and consuming 400 mg (about half the weight of a small paper clip) of calcium and 500 IU of vitamin D daily. The trial also had the following exclusion criteria: (1) those with metabolic bone disorders, renal disease, cancer, cardiovascular disease, diabetes mellitus, respiratory disease, gastrointestinal disease, liver disease, or other chronic diseases, (2) heavy smokers (more than 20 cigarettes per day), (3) perimenopausal women with a chance or plan of pregnancy, (4) taking prescription medications that can alter bone and calcium metabolism, (5) using anabolic agents, and (6) consuming more than 2 alcoholic beverages per day. Baseline

data were collected before participants consumed any blackcurrant supplementation and utilized for this study.

### 2.2. Data Collection

Medical histories and sociodemographic histories were collected at an initial interview (the timeline of collection is displayed in Figure 1). During the baseline visit, dietary data were collected from a three-day food record including all foods and beverages consumed during two non-consecutive weekdays and one weekend day. The nutrient intake was analyzed via the Nutrition Data System for Research (NDSR, University of Minnesota Nutrition Coordinating Center, Minneapolis, MN). Physical activity data were also collected at baseline from a 1-week record and analyzed by calculating the metabolic equivalent of task (MET) score [26]. In addition, the baseline visit included a physical exam to determine weight (kg), height (m), BMI (kg/m$^2$), waist circumference (cm), and a complete DXA scan at the University of Connecticut Korey Stringer Institute to determine whole-body, head, arms, legs, trunk, ribs, spine, and pelvis BMD along with body composition.

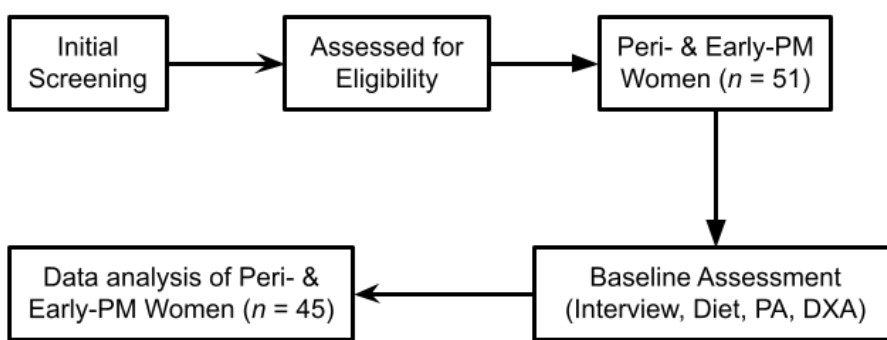

**Figure 1.** Description of data collection protocol. PM, postmenopausal; PA, physical activity; DXA, dual X-ray absorptiometry.

### 2.3. Data Analysis

To analyze the data, participants were split into two groups based on whole-body BMD data having a Z-score less than 0 or a Z-score greater than or equal to 0. A Z-score compares a person's bone density to the average bone density of people their own age and gender. Thus, having a Z-score greater than or equal to 0 means that the person's bone density is the same or higher than that of others of the same age, sex, and body size [27]. All participants in the present study, except for three, had a Z-score greater than −1. Thus, to create groups for the statistical analysis, a Z-score of 0 was chosen as the cutoff for low BMD in the current study population. To examine the differences in body composition, physical activity, and dietary intake between two groups, we utilized a *t*-test for continuous variables and a $\chi^2$ test for categorical variables.

To analyze the percentage of participants meeting nutrition recommendations, participants daily intake was compared to Adequate Intake (AI) of potassium (2600 mg/day), Recommended Dietary Allowance (RDA) of vitamin E (15 mg/day), and the Dietary Guidelines of America (DGA) recommendation of less than 10% of total calories from added sugar per day [28–30]. A $\chi^2$ test was then used to assess the association between BMD and the number of individuals meeting each recommendation (greater than or equal to the recommendation).

Continuous variables are expressed as means and standard deviations, and categorical variables are shown as numbers and percentages. Differences with $p < 0.05$ were considered significant. Pearson's correlation analysis was used to assess the association between total calorie intake, nutrient intake status related to osteoporosis, and bone density. Pearson's correlation analysis was performed using SAS (Version 9.4, SAS Institute, Cary, NC, USA), and $p < 0.05$ was considered statistically significant.

## 3. Results

### 3.1. Demographic Characteristics of Participants

Displayed in Table 1 are the demographic characteristics of all 45 participants. Most participants had a college degree, were Caucasian, and had a household income above USD 90,000. There were no significant differences in demographic factors between the two groups aside from whole-body BMD.

**Table 1.** Demographic characteristics of participants.

| Factor | All Participants (*n* = 45) | Group 1, Z < 0.0 (*n* = 17) | Group 2, Z ≥ 0.0 (*n* = 28) | *p*-Value (*t*-Test) [a] |
|---|---|---|---|---|
| Whole-body BMD (Z) | $0.35 \pm 1.06$ | $-0.74 \pm 0.56$ | $1.03 \pm 0.64$ | <0.05 |
| Household annual income [b] | | | | |
| <USD 90,000 | 10 (23.26) | 2 (11.76) | 8 (30.77) | 0.607 |
| USD 90,000–USD 125,000 | 13 (30.23) | 7 (41.18) | 6 (23.08) | |
| >USD 125,000 | 20 (46.51) | 8 (47.06) | 12 (46.15) | |
| Education [b] | | | | |
| Has college degree | 34 (79.07) | 14 (82.35) | 20 (76.92) | 0.669 |
| No college degree | 9 (20.93) | 3 (17.65) | 6 (23.08) | |
| Race/ethnicity [b] | | | | |
| White | 39 (90.70) | 15 (88.24) | 24 (92.31) | 0.130 |
| Hispanic | 1 (2.33) | 0 (0) | 1 (3.85) | |
| Asian | 2 (4.65) | 1 (5.88) | 1 (3.85) | |
| Other | 1 (2.33) | 1 (5.88) | 0 (0) | |

Values displayed as means ± SD or *n* (%). [a] *t*-test between Groups 1 and 2. [b] Analysis was performed on 43 participants for household income, education, and race/ethnicity due to missing data.

### 3.2. Body Composition Factors of Participants and Bone Mineral Density

Various factors of body composition and fat distribution were compared between the two groups, and the results are depicted in Figures 2 and 3. Many body composition factors were significantly greater with BMD Z-scores above zero, including BMI ($p < 0.01$), lean mass ($p < 0.01$), fat mass ($p < 0.01$), android fat ($p < 0.05$), ratio of trunk to limb fat mass ($p < 0.05$), relative skeletal muscle index (RSMI) ($p < 0.01$), and RMR ($p < 0.01$).

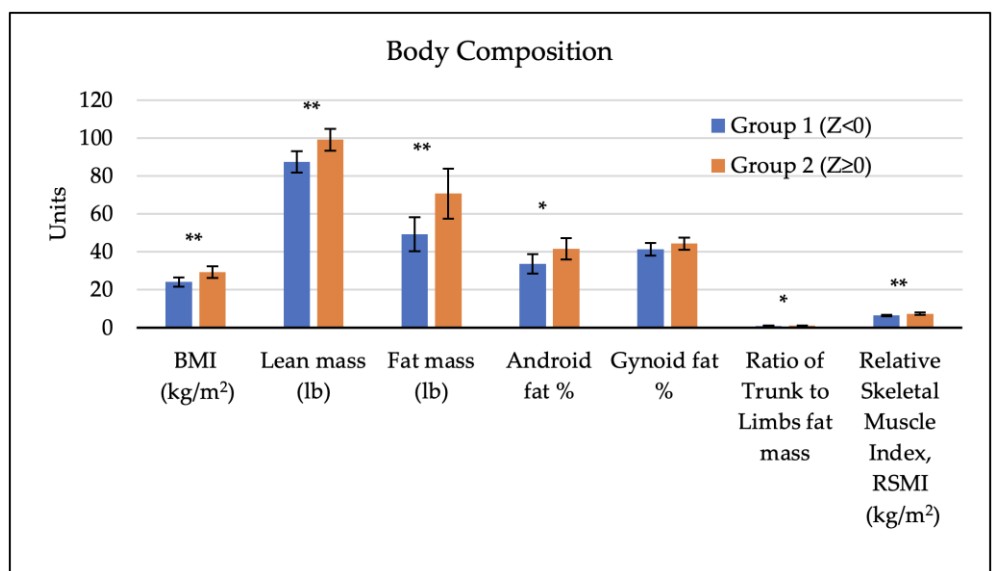

**Figure 2.** Body composition factors and BMD among peri- and early postmenopausal women. Bars represent means while error bars represent standard deviation. * $p < 0.05$; ** $p < 0.01$.

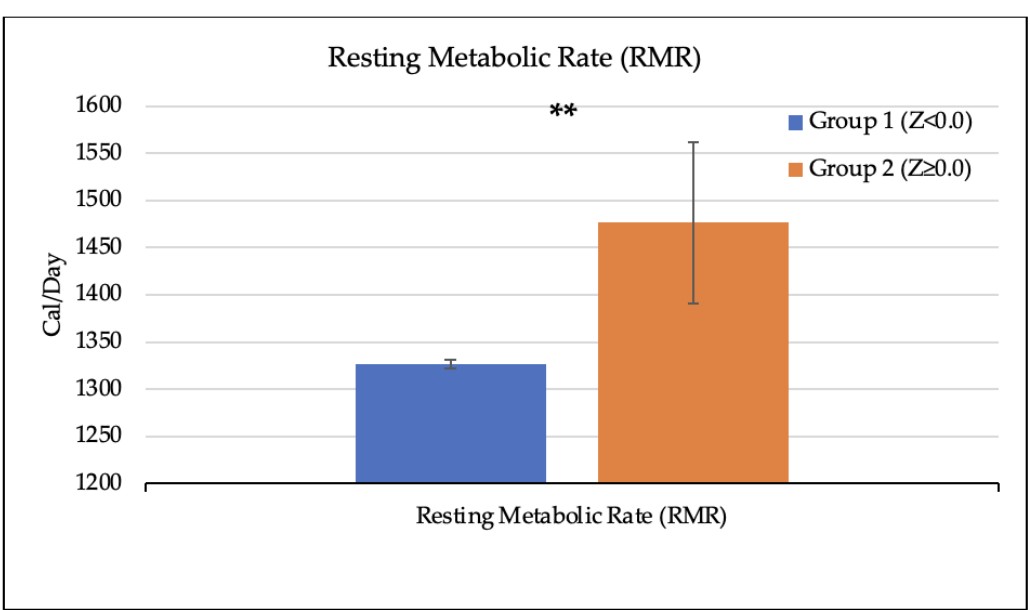

**Figure 3.** Resting metabolic rate (RMR) and BMD among participants. Bars represent means, while error bars represent standard deviation. ** $p < 0.01$.

### 3.3. Physical Activity of Participants and Bone Mineral Density

Using MET scores, the average kilocalories of daily physical activity expenditure of participants were estimated for total physical activity, leisure time physical activity, and occupational physical activity, and the results are shown in Figure 4. Total physical activity and occupational physical activity were significantly higher in Group 2 than in Group 1 ($p < 0.05$).

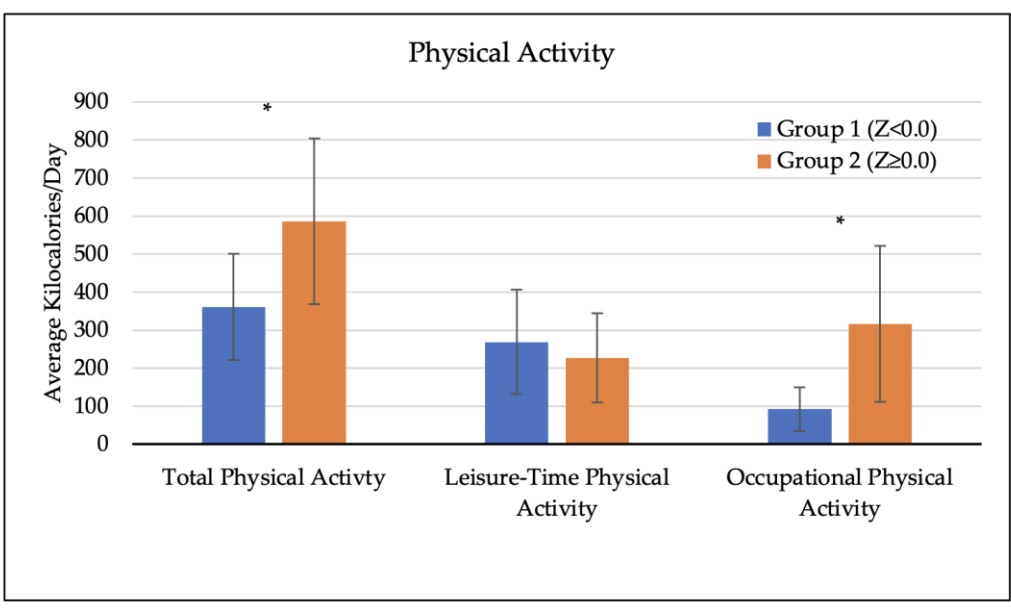

**Figure 4.** Physical activity of participants and BMD. Bars represent means, while error bars represent standard deviation. * $p < 0.05$.

### 3.4. Dietary Factors of Participants and Bone Mineral Density

Figure 5 shows the results of nutrition recommendation guidelines among both groups. The AI, RDA or DGA was analyzed in association with BMD. The percentage of women consuming more than or equal to the potassium recommendations (AI, 2600 mg/day) was

significantly greater with positive BMD Z-scores ($p < 0.05$), and the number of women who consumed adequate vitamin E (RDA, 13 mg/day) was significantly greater with low BMD Z-scores ($p < 0.05$). Percentage of people whose added sugar intake met the recommended guideline (DGA, <10% total calories from sugar) was higher in group 2 than in group 1; however, this trend was not significantly significant.

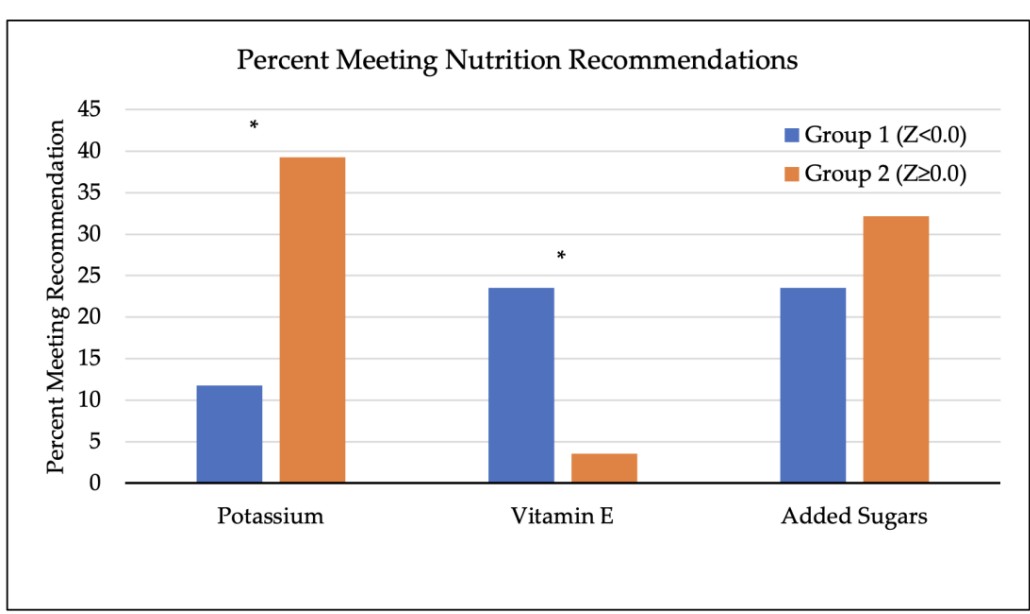

**Figure 5.** Adequacy of nutrient intake and BMD. * $p < 0.05$. AI of Potassium, 2600 mg/day; RDA for Vitamin E, 15 mg/day; DGA for added sugar, <10% total daily calorie intake.

Figure 6 shows the results of certain food groups that differed by BMD Z-score. Participants with higher whole-body BMD had greater intakes of alcohol and trans fatty acids ($p < 0.05$).

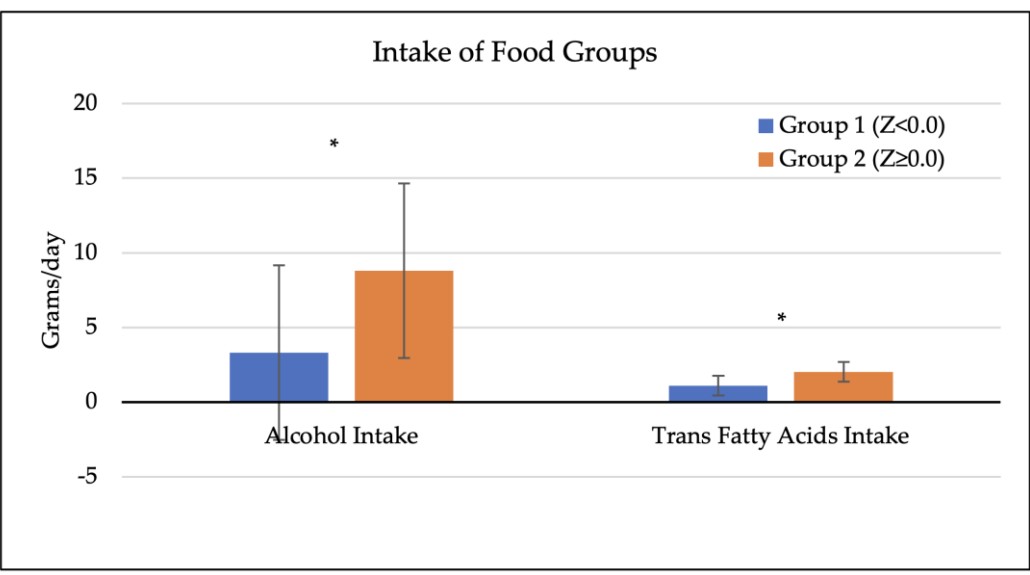

**Figure 6.** Intake of food groups and BMD. Bars represent means, while error bars represent standard deviation. * $p < 0.05$.

*3.5. Correlations between Total Calorie Intake, Nutrient Intake Status Related to Osteoporosis, and BMD*

Table 2 lists the correlations of whole-body BMD, total calorie intake, and status of nutrient intake (including nutrients related to osteoporosis) with one another. Total calorie intake was significantly positively correlated with added sugar, vitamin E, potassium, alcohol, trans fatty acid, calcium, and vitamin D intakes ($p < 0.05$). Whole-body BMD was positively correlated with added sugars ($p < 0.05$). There was a positive correlation between vitamin D intake and vitamin E, potassium, trans fatty acid, and calcium intakes ($p < 0.05$). Calcium intake was positively correlated with added sugars, vitamin E, potassium, and trans fatty acids ($p < 0.05$). Trans fatty acid intake was also positively correlated with alcohol ($p < 0.05$).

**Table 2.** Correlations between total calorie intake, nutrient intake status, and bone density in peri- and early postmenopausal women.

| | Whole-Body BMD | Total Calories | Added Sugars | Vitamin E | Potassium | Alcohol | Trans Fatty Acids | Calcium | Vitamin D |
|---|---|---|---|---|---|---|---|---|---|
| Whole-Body BMD | 1.0 | | | | | | | | |
| Total calories | 0.20579 (0.1802) | 1.0 | | | | | | | |
| Added sugars | 0.30032 (0.0476) * | 0.37379 (0.0124) * | 1.0 | | | | | | |
| Vitamin E | −0.02822 (0.8557) | 0.45351 (0.0020) * | 0.03309 (0.8312) | 1.0 | | | | | |
| Potassium | 0.09037 (0.5596) | 0.50223 (0.0005) * | 0.00111 (0.9943) | 0.67198 (<0.0001) * | 1.0 | | | | |
| Alcohol | 0.13251 (0.3912) | 0.50247 (0.0005) * | −0.10197 (0.5101) | 0.03184 (0.8374) | 0.12593 (0.4153) | 1.0 | | | |
| Trans fatty acids | 0.16705 (0.2784) | 0.49853 (0.0006) * | 0.06899 (0.6563) | −0.09928 (0.5214) | 0.04174 (0.7879) | 0.56243 (<0.0001) * | 1.0 | | |
| Calcium | 0.21885 (0.1535) | 0.52041 (0.0003) * | 0.39916 (0.0073) * | 0.46289 (0.0016) * | 0.41897 (0.0046) * | 0.19362 (0.2079) | 0.31577 (0.0368) * | 1.0 | |
| Vitamin D | 0.02204 (0.8871) | 0.74634 (<0.0001) * | 0.05598 (0.7182) | 0.57478 (<0.0001) * | 0.46271 (0.0016) * | 0.26600 (0.0809) | 0.33201 (0.0277) * | 0.47527 (0.0011) * | 1.0 |

Values displayed as correlation coefficients (*p*-values). * $p < 0.05$.

## 4. Discussion

The main objective of the current study was to assess the differences in sociodemographic, body composition, physical activity, and dietary factors in peri- and early postmenopausal women with low and high whole-body BMD Z-scores. Women with positive BMD Z-scores had significantly higher body composition factors, total and occupational physical activity, intakes of macronutrient components, and adequacy of some dietary groups, with the exception vitamin E, compared with women with negative BMD Z-scores.

The body composition results of the current study support the findings of the previous literature discussing the associations of BMD with BMI [7,8], lean mass [9], fat mass [9], resting metabolic rate (RMR) [9], android fat [10], ratio of trunk fat mass to limb fat mass [11], and relative skeletal muscle index in postmenopausal women [12]. Our study showed significantly higher BMIs in women with greater BMDs (Z-scores ≥ 0) than women with lower BMDs (Z-scores < 0). Similarly, previous studies found a statistically significant positive association between BMI and femur bone density in postmenopausal women [7] and a positive association between BMI and BMD [8]. Additionally, previous reports indicate that BMI may be a contributing factor for BMD, as increased body mass often leads to increased levels of hormones, such as estrone and estradiol, associated with bone building [31]. Our study further found that lean and fat mass were significantly higher in women with positive BMD Z-scores compared with negative BMD Z-scores. This was

similarly found in a study population of women within 5 years post-menopause that identified an association between lean mass and BMD, with lean mass being a potential BMD predictor [9]. The study also found a significant positive association between fat mass and BMD in women. Previous reports have supported varying associations between lean mass, fat mass, and BMD [9,32,33]. The relationship of lean and fat mass with BMD may be explained due to the function of mass in mechanical loading, aiding in the development of bone. Lean mass, which is the weight of the body minus fat tissue, may be associated with bone density due to the influence of one of its main components, muscles. Increased muscle mass and force from muscles can aid in bone development [32]. Higher levels of fat mass (weight of fat tissue in the body) and adipose tissue result in higher levels of hormones like estrone and leptin, which can increase the activity of osteoblasts, cells that function to build bone, and decrease the activity of osteoclasts, cells that resorb bone [33,34]. In addition, fat mass can increase the weight and force experienced by bones, which can lead to increased BMD [34]. Furthermore, a previous study reported a positive association between RMR and BMD [9]; similarly, our study found that RMR is higher in peri- and early postmenopausal women with positive BMD Z-scores than those with negative Z-scores. As RMR is correlated with lean body mass and lean body mass is a predictor for BMD, RMR may be a predictor for BMD in peri- and postmenopausal women. RMR is also affected by physical activity and, thus, may be a predictor for BMD based on the association between physical activity and BMD [35].

As mentioned in previous studies, much research has been conducted on the association between fat mass and BMD, but the associations of regional android and gynoid fat with BMD have been studied less. Our study supported that the percentage of android fat (percentage of total body fat concentrated around the abdominal and upper body regions), but not the percentage of gynoid fat (fat around the hip and thigh regions of the body), is significantly higher in women with positive BMD Z-scores [10,36]. This finding supports a previous study that identified a positive association of android fat with BMD in postmenopausal women [10]. This association may be explained by increased levels of adipocytes within android fat and hormone secretion that can aid in increasing bone buildup and decreasing bone breakdown, supporting bone density. However, the mentioned study also found an association between gynoid fat and BMD that we did not observe in the current study [10]. Additionally, our study supported a positive association between the ratio of trunk fat mass (fat in the chest, abdomen, and pelvic body regions) to limb fat mass (fat in the arms and legs) and BMD Z-scores in participants. Our study reported a higher ratio with increased BMD, meaning that participants with fat concentrated around their trunks over their limbs had higher bone densities. Previously, it has been reported that trunk fat mass was positively associated with BMD, but not limbs or peripheral fat mass, in pre- and postmenopausal women [11]. Research has suggested that trunk versus peripheral fat mass may have differing effects on metabolic disorders, such as insulin resistance and hyperlipidemia [36,37]. A previous study reported a higher risk of osteoporosis in women with increased insulin resistance, with a potential mechanism of increased inflammation and oxidative stress resulting in bone loss [38]. Reports also indicate a positive association between hyperlipidemia and osteoporosis, as high cholesterol can result in the activity of osteoclasts increasing and osteoblasts decreasing, resulting in bone loss [39]. Our study found a significantly higher RSMI in women with positive BMD Z-scores. RSMI is a measure that can be used to assess the mass of skeletal muscles in the body. A previous study did not find an association between skeletal muscle index and BMD in postmenopausal women [40]; however, other literature has reported an association between skeletal muscle mass and bone mass [12]. Skeletal muscle mass is a component of our musculoskeletal system that may have an impact on osteoporosis, as skeletal muscles are important in supporting levels of BMD with their mechanical loading [12]. Thus, as the RSMI indicates the relative mass of muscle in the body's limbs, an increased RSMI may be associated with increased BMD due to these effects on mechanical loading. The various findings between studies may be due to the populations studied, study methods utilized,

weight and adiposity, time in menopause, and muscle mass indexes, as these can affect bone mass.

The current study reported that physical activity and occupational physical activity are significantly higher in peri- and early postmenopausal women with positive BMD Z-scores than those with negative Z-scores. This finding is in line with the previous literature reporting that physical activity may help prevent osteoporosis by reducing bone loss, specifically with a long-term exercise program, daily moderate physical activity, and a multicomponent activity program [14]. Physical activity, in general, can increase forces on bone, increasing bone remodeling and building, especially in areas that have a higher impact [14–16]. In addition, physical activity can stimulate osteoblast activity while promoting circulation, which can increase bone metabolism and build-up [41]. However, the current study did not observe that leisure time physical activity was significantly higher in women with positive BMD Z-scores, and, thus, physical activity from occupational sources may contribute to the preventive associations identified.

This study further identified associations with various macronutrient components and adequacy of food groups, vitamins, and minerals with BMD in peri- and early post-menopausal women. There was a higher intake of alcohol and trans fatty acids observed in women with positive BMD Z-scores versus negative Z-scores. These findings support a previous study's report that moderate alcohol intake was associated with higher BMD [42]. Alcohol has a complex relationship with bone, as some studies report that it can increase bone loss at high levels, but there is a moderate number that suggests it supports bone health by supporting levels of hormones such as estradiol [43]. Regarding trans fatty acids, previous studies support an inverse relationship between plasma trans fatty acids and BMD, which conflicts with our findings. The mentioned study assessed plasma trans fatty acids among all adults, including men and women of varying age levels [44]. Thus, there is a potential that our findings differ due to the sample size, sample population, and methods of trans fatty acids analysis, as our study utilized dietary data, while the mentioned study utilized plasma levels. Another report discussed how conjugated linoleic acid may be beneficial in decreasing the activity of osteoclasts, thereby preventing bone loss and supporting bone health [45]. Our study also identified a positive correlation between trans fatty acid intake and alcohol intake, which may support their actions to promote bone health. A previous study also reported an inverse association between sugars (in terms of sugar-sweetened beverages) with BMD [46], whereas we did not observe an association with added sugar. The different findings may be that the present study assessed added sugars from all sources, while the aforementioned study utilized added sugar intake only from sugar-sweetened beverages. In addition, our study focused on peri- and early postmenopausal women, while the previous study assessed all adults. Previous reports suggested that mice consuming high-sugar diets had increased storage of calcium, increased osteoblast activity, and decreased osteoclast activity that was associated with increased bone density [47]. Added sugars were positively correlated with calcium and total calorie intake. Calcium has been reported to be associated with higher bone density in postmenopausal women [17,18], and those consuming more calories may be consuming more added sugars overall.

Our study additionally observed that women with positive whole-body BMD Z-scores had lower adequacy of vitamin E intake. A previous study reported that vitamin E supplementation, commonly in the alpha-tocopherol form, may result in increased bone loss in postmenopausal women by decreasing levels of gamma-tocopherol, a compound that increases bone development by decreasing bone resorption [48]. Vitamin E was positively correlated with total calorie intake, vitamin D intake, and calcium intake. Vitamin D and calcium are nutrients that lead to increased bone density, and, thus, this finding may underscore a conflicting relationship between vitamin E and bone loss [17,18]. We also observed higher adequacy of potassium intake in women with positive BMD Z-scores. Another study reported that potassium intake was associated with increased bone health and a decreased risk of osteoporosis [49]. Potassium can help support bone density by

decreasing bone resorption due to maintaining pH and increasing calcium absorption in the kidneys [49]. Potassium was positively correlated with total calorie, vitamin D, and calcium intakes. Due to vitamin D and calcium's effect of decreasing bone loss, this may support potassium's role in decreasing the risk of osteoporosis [17,18]. We additionally found that total calorie intake was positively associated with vitamin D, calcium, and all nutrients that had associations with bone density in this study, potentially due to increased overall consumption of calories including more nutrient intake. However, total calorie intake was not associated with bone density, and, thus, the associations found between the nutrient factors and bone density may be due to the nutrients' effects.

## 5. Study Limitations

The findings of this study should be interpreted with attention to some limitations. First is the sample size, which was small. With a larger sample size, we could have greater power to detect associations between risk factors and BMD. Additionally, our study population included mainly healthy individuals, and thus the body composition, physical activity, and nutrient intake factors could not be assessed at various BMD levels attributed to osteoporosis. Third, the study population lacked diversity, including mostly Caucasian women with an upper-middle socioeconomic status in a narrow age range (45–60 years), thus affecting the generalizability. Fourth is the cross-sectional study design, and, therefore, causation cannot be inferred. The current study has several strengths, including that detailed dietary and physical activity data were able to be collected with dietary and physical activity reports over multiple days, aiming to identify usual intake and activity. Additionally, the target population was in menopausal transition, and, thus, the results can be beneficial to identify preventative strategies.

## 6. Conclusions

Overall, this study supports that there are differences in body composition factors, physical activity, especially occupational physical activity, and certain nutrient components in peri- and early postmenopausal women with positive whole-body BMD Z-scores compared with negative BMD Z-scores. The results warrant further investigation with a larger sample size and broader age range and demographics that are more representative of the general population to identify preventative factors of osteoporosis. Additionally, further studies may be carried out with an intervention trial aimed at clarifying the relationships between specific physical activity regimens and bone health.

**Author Contributions:** Conceptualization, O.K.C. and D.B.P.; methodology, D.B.P., B.M.N. and M.D.M.; resources, O.K.C.; formal analysis, D.B.P., B.M.N. and M.D.M.; investigation, D.B.P., B.M.N. and M.D.M.; data curation, D.B.P., B.M.N. and M.D.M.; writing—original draft preparation, D.B.P., B.M.N. and O.K.C.; writing—review and editing, D.B.P., B.M.N., M.D.M. and O.K.C.; supervision, O.K.C.; project administration, O.K.C.; funding acquisition, O.K.C. All authors have read and agreed to the published version of the manuscript.

**Funding:** This research was funded by the USDA NIFA Seed Grant (#2020-67018-30852) to Dr. Ock K. Chun.

**Institutional Review Board Statement:** This study was conducted in accordance with the Declaration of Helsinki and approved by the Institutional Review Board of the University of Connecticut (protocol code HR20-0035 and 4 June 2020).

**Informed Consent Statement:** Informed consent was obtained from all subjects involved in this study.

**Data Availability Statement:** Data is contained within the article.

**Acknowledgments:** We would like to acknowledge the efforts of all the study participants who volunteered to participate, especially during the difficult and trying period of the COVID-19 pandemic.

**Conflicts of Interest:** The authors declare no conflicts of interest.

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
