# Peer review of "Identification of the Risk Factors Associated with Low Bone Density in Peri- and Early Postmenopausal Women"

_2674-0311, doi:10.3390/dietetics3010007_

Round 1
Reviewer 1 Report
Comments and Suggestions for Authors
The authors focus on the identification of the risk factors associated with estrogen deficiency-induced bone loss in peri- and early postmenopausal women. The study is very interesting, and the manuscript is well-written. The study respects scientific principles and is based on appropriate protocol. The discussion is well conducted. References are typical and actual. However, the work has some shortcomings that should be filled and some errors to be corrected.
- There is no description of group 2 in Figure 2
- there are no error bars in Figure 5
-Please add Study Limitation as a separate section
Reviewer 2 Report
Comments and Suggestions for Authors
Title : Identification of the Risk Factors Associated with Estrogen Deficiency-Induced Bone Loss in Peri- and Early Postmenopausal Women
Research related to this should be oriented toward cross-sectional studies targeting postmenopausal women. In order to improve the completeness of the study, I would like to suggest several modifications.
line 1-2; The title does not match the author's research. In this cross-sectional study, the effects of blackcurrant (BC) supplementation on osteoporosis were investigated in postmenopausal women. Accurate titles are required to convey the author's research to readers at once.
line 45-46 ; Bone health has been found to be associated with multiple factors in postmenopausal women, including body composition, physical activity and dietary factors [7-23].
It seems that 17 references in one sentence do not provide an accurate guideline. Do you want your readers to refer to 17 papers to rationally understand this one sentence? What are the exact references for readers?
line 71-73 and line 78-79 ; Please compare the contents of the two paragraphs and then revise the contents. The purpose of this study and the actual research design are not consistent. If this study was designed to verify the effectiveness of BC, the research aim in the introduction should be revised.
line 76-77; In the introduction, the author developed logic about postmenopausal women and osteoporosis. However, in the study design, peri-menopausal women and early postmenopausal women were selected as subjects. The background for selecting these two groups as subjects for osteoporosis research should be explained, and the E2 (Estradiol) level should be presented as a clear way to distinguish between the two groups. There is a lack of uniformity in the introduction and research implementation.
Line 80-82; In the study design, Women were recruited from Northeastern Connecticut and were randomized into one of three groups: (1) Control (one placebo capsule/day), (2) Low BC (392 mg BC/day) or (3) High BC (784 mg BC/day). However, all results are presented in groups 1 and 2. What does this manuscript want to explain? As you can see, the content of the research design and the research title are not in harmony.
Line 111-119; In the study design, the BC diet was administered in three groups. Why is the data analysis presented in only 2 groups? You must add explanations that your readers will understand.
Line 97-99; During the baseline visit, dietary data was collected from a 3-day food record including all foods and beverages consumed during two non-consecutive weekdays and one weekend day.
-In addition to adding analysis of total calories and nutrients related to osteoporosis, the results should show correlation analysis between total calorie intake status by group, nutrient intake status related to osteoporosis, and bone density. For example, calcium and vitamin D.
-Please present analysis data on the rationality of setting markers such as vitamin E that were analyzed in the research results and their relationship with osteoporosis.
- An analysis of why the blackcurrant-related part was selected in this study and how it affects the results should be presented. Does it make sense to design and write this study to present the effects of blackcurrants on bone mineral density?
- There are three types of estrogen (estrone, estradiol, and estriol). Please add clarity in your terminology throughout the paper to clarify what type of Estrogen you mean.
- In introduction, authors described that "Thus, the aim of this study was to investigate how various body composition, physical activity, and dietary factors are associated with bone loss in peri- and early postmenopausal women." However, in the main text, data unrelated to the research goal ex; education level, annual income, race, etc. are described. If you want to add these data, I think it would be appropriate to mention these in introduction and present the data in the result.
- In Figure 1, the authors suggested that blood was collected from the subject. However, the manuscript does not provide any blood-based indicators. There is no justification for recording blood sampling if the data is not presented. I recommend revising this part.
Comments on the Quality of English Language
In statistical display, p value notation is not unified. Lowercase letters must be written in italics. (ex: line 133-136). Other English expressions seem to have no problems.
Reviewer 3 Report
Comments and Suggestions for Authors
Dear authors,
the manuscript presented to me for evaluation - judging by the title, it looked promising. Unfortunately, many mistakes were made that affected its quality. Please correct the manuscript according to the recommendations, then it will be able to be published.
1. On what basis did the authors divide patients into: (BMD) Z-score greater than and less than zero. The cited literature does not explain this division. please explain.
2. The title of the manuscript suggests that the analyzes will also cover estrogen, but this is omitted. The problem of estrogen deficiency was only mentioned a few times throughout the manuscript. please clarify and change the title or add estrogen analysis to the manuscript.
3. In the materials and methods, the authors divided the study group into 4, and later write about two, completely different groups. Please explain.
4. In the manuscript, the authors mentioned blackcurrant supplementation - but they did not continue this further. I feel like this is part of another manuscript. Please explain.
6. In Figure 2, part of the legend is missing.
7. there are no standard deviations in Figure 5.
Round 2
Reviewer 2 Report
Comments and Suggestions for Authors
The authors faithfully revised the manuscript according to the reviewer's suggestions.
Reviewer 3 Report
Comments and Suggestions for Authors
Dear authors,
Thank you for correcting the manuscript. The manuscript should be published in its current form.
I only have a small comment on Fig. 2. There is no need to repeat the title and caption of the figures "Description of data collection protocol".
Reviewer